# ZnO Nanoparticles for Photocatalytic Application in Alkali-Activated Materials

**DOI:** 10.3390/molecules25235519

**Published:** 2020-11-25

**Authors:** Hector R. Guzmán-Carrillo, Alejandro Manzano-Ramírez, Ines Garcia Lodeiro, Ana Fernández-Jiménez

**Affiliations:** 1CINVESTAV-I.P.N. Unidad Querétaro, Querétaro 76230, Mexico; amanzano@cinvestav.mx; 2Eduardo Torroja Institute of Construction Sciences (IETcc–C.S.I.C.), E28033 Madrid, Spain; iglodeiro@ietcc.csic.es

**Keywords:** composites, chemical synthesis, electron microscopy, X-ray diffraction, photocatalytic properties

## Abstract

This paper reports an Alkali-Activated Materials (AAM) using two different precursors, metakaolin and a metallurgical slag with photocatalytic zinc oxide nanoparticles, as novel photocatalytic composites. The photodegradation performance of the composites using methylene blue (MB) dye as a wastewater model was investigated by ultraviolet radiations (UV-vis) spectroscopy. Adsorption in dark conditions and photodegradation under UV irradiation are the mechanisms for removing MB dye. The pseudo-first-order kinetic and pseudo-second-order kinetic models were employed, and the experimental data agreed with the pseudo-second-order model in both cases with UV and without UV irradiations. As new photocatalytic materials, these composites offer an alternative for environmental applications.

## 1. Introduction

Alkali activation of solid aluminosilicates materials (AAM) is a technology that has shown the potential to make green cement, also known as alkali-activated cement (AAC). This material began to develop in the 20th century [1]. However, in the 21st century, these materials have gained importance due to their lower CO_2_ emissions than the Ordinary Portland Cement (OPC). The OPC industry produces approximately one ton of CO_2_ for each ton of cement manufactured, and it is well-known that CO_2_ is one of the main gases responsible for global warming. For that reason, AAM with a low footprint is a real alternative to the traditional OPC-based materials. Another critical issue is industrial waste disposal, i.e., fly ash, bottom ash, metallurgical slag, rice husks, and the valorisation of these by-products through alkali-activation is feasible.

The nanostructure of AAM, strongly depends mainly on the calcium content in the precursors (raw materials). With high calcium content of, i.e., metallurgical slags, the main reaction product is a calcium-aluminium-silicate hydrated gel (also known as C-A-S-H gel) where silicon forms large networks [2]. Meanwhile, in systems with low calcium content, the product formed is an alkali aluminosilicate hydrated gel (N-A-S-H gel) with three-dimensional structures as a zeolitic type [3].

As AAM is highly porous, its use as an absorbent for removal of heavy metals like lead [4], copper, nickel, zinc [5], and other toxic compounds present in residual waters like dyes by industrial discharges [6,7], have been investigated.

The removal of dyes in wastewater could be achieved by different means—absorption, chemical degradation, among others—but the search for new, simpler, and more economical alternatives for the complete elimination of these types of contaminants is not an easy task. The absorption process offers an economical and easy way to partially eliminate the contaminants, but a combination with other techniques is required. Heterogeneous photocatalysis comprises the absorption of a photon for a semiconductor to form a pair electron-hole on the surface, forming highly reactive species that can degrade a wide variety of contaminants present in residual waters [8].

Different nanoparticles (NPs) have been investigated as potential photocatalytic materials such as TiO_2_, ZrO_2_, V_2_O_5_, Fe_2_O_3_, ZnO, and Cu_2_O. The widest nanoparticle studied is TiO_2_ [9,10,11,12,13]. Nevertheless, nanostructured zinc oxide (ZnO) is attractive as a catalyser for its wide direct bandgap, highly electronic mobility, no toxicity, and facile control of morphology [14].

Particles with nanometric sizes have different chemical and physical properties compared with their bulk counterparts. It is hoped that nanomaterials used like catalysers present better performance due to a higher surface area and a change in the properties for defects on their surface. Alkali-activated materials with photocatalytic activity with the addition of TiO_2_ [15], Cu_2_O [16], and Cu_2_O/TiO_2_ nanoparticles [17] have been previously reported.

In this paper, not only the re-use of industrial wastes but also the functionalization of AAM with photocatalytic nanoparticles of zinc oxide was investigated. The use of ZnO in AAM is a new and interesting research line. In the literature, only a few works [18] have addressed this possibility with promising results. However, there are still knowledge gaps to be resolved. This work aims to increase the knowledge on this particular subject.

## 2. Results

### 2.1. Characterization

Figure 1 shows the diffractogram of the starting Metakaolin (MK) and of the alkali-activated pastes at 28 days: without nanoparticles (mkW), with nanospheres (mkS) and nanorods (mkR) without functionalizing, and functionalized (mkSf and mkRf respectively). The raw MK shows an amorphous hump at 2θ between 20° and 30° and some crystalline phases such as cristobalite C (ICDD-PDF#39-1425 file) and α-quartz Q (ICDD-PDF#46-1045 file).

Apparently, there is no difference between X-ray diffraction (XRD) patterns of anhydrous MK and metakaolin-based pastes. Still, with an exhaustive analysis, it is possible to observe a shift of the halo towards higher 2θ values, usually associated with the precipitation of the amorphous phase of the N-A-S-H type gel [19]. In the pastes doped with the nanoparticles (mkS, mKR, mKSf, mKRf), additional peaks indicating the ZnO phase’s presence were detected. This shows that the ZnO particles have been incorporated into the pastes and apparently have not been decomposed or have chemically reacted, forming other reaction products.

Figure 2 shows the XRD diffractograms of the precursor SL and the SL pastes, without nanoparticles (slW), and with nanoparticles (slS, slR, slSf, slRf, nanospheres, nanorods without and with functionalized, respectively) at 28 days. The XRD patterns of the metallurgical slag exhibited a significant hump at 2θ values of 38° to 40° associated with the amorphous component that it consisted primarily of a vitreous phase of slag, together some crystalline phases as merwinite (M) (ICCD-PDF#35-0591 file), and calcite (T) (ICCD-PDF#47-1743 file). The patterns for pastes showed a slight shift in the hump, denoting a reaction of the SL with the activator and the formation of a cementitious gels type C-A-S-H [20]. Also, a crystalline phase hydrotalcite, H (ICCD-PDF#41-1428 file), was detected [21,22]. As in MK-pastes, additional peaks indicating the presence of ZnO phase were detected in the diffractograms of slS, slR, slSf and slRf pastes. ZnO particles have been incorporated into the pastes and have presumably not reacted.

SL-bases paste doped with functionalized ZnO NPs lead to some changes in the microstructure, showing an increase in the compactness, as shown in Figure 3. On the contrary, in the MK-based pastes, no significant differences in the microstructures are observed.

Figure 4 shows an elemental mapping (by EDS) of the pastes with nanospheres modified with CMC (mkSf). It is possible to observe an excellent homogeneous distribution of ZnO NPs in the paste. This result leads to expect a good photocatalytic performance of this sample since the nanoparticles do not react with the paste. On the other hand, Figure 5 shows that the nanospheres with CMC in slag-based paste form some kind aggregates and a non-homogeneous distribution of nanoparticles is observed.

The chemical analysis shows considerably lower detected values of ZnO compared with the 10-percentage added. This would suggest that in a slower geopolymerization kinetics, i.e., SL pastes, the nanoparticles are exposed to a high pH exposition during more time, which leads to the breakdown of ZnO nanoparticles since ZnO is an amphoteric material; therefore, it can dissolve in alkaline solutions, as shown in Equations (1) and (2) [18].
(1)$$Acid\;media:ZnO+\;2H^{+}\;\to \;Zn^{2+}+\;H_{2}O\;$$
(2)$$Basic\;media:ZnO+\;H_{2}O+\;2OH^{-}\;\to \;{\left[Zn{\left(OH\right)}_{4}\right]}^{2-}\;$$

Meanwhile, when the geopolymerization process is faster, i.e., MK pastes, ZnO nanostructures’ contact in basic pH is shorter. The stabilization is better for the functionalization with CMC; for that reason, the chemical analysis shows similar values than the percentage added.

To determine the photocatalytic performance of the pastes with nanoparticles, and how these interact with the matrix, the photocatalytic effect will be analysed using a cationic dye as model pollutant dye, methylene blue. The cationic dye, MB, was chosen due to its reflected adsorptive capacity, which is higher than other organic dyes [23]. It is worth mentioning the relationship between the adsorption and the photocatalytic activity since the organic dyes’ photodegradation reaction occurs after adsorbed on the surface [24]. As a result, adsorption contributes simultaneously to accelerate photocatalytic action and participate in removing dyes. The MB colour is also stable at the high pH of the AAM [17]. Hence, in the present work, MB is considered suitable to evaluate the composites’ photodegradation performance.

### 2.2. Photocatalytic Activity

As was mentioned before, to determine the adsorption equilibrium time, the experiments were conducted in dark conditions, and the results are shown in Figure 6a and Figure 7a. The residual concentration of MB is plotted for the different metakaolin-based pastes. According to the results, it is possible to observe an equilibrium sorption-desorption after only 30 min for the paste without nanoparticles and the rest of the samples follow a similar behaviour, Figure 6a. In dark conditions, during the first minutes, a rapid reduction of MB concentration is related to the availability of active sites on the surface that in time will decrease to reach stability. It should be noticed that for the paste without nanoparticles in dark conditions, the residual concentration of MB has a similar behaviour under UV light conditions, there is a small decrease related to the effect of the lamp, photolysis. When the lamp is on after the first 30 min, there is a change in residual concentration due to photoactivation of the zinc oxide nanostructures on the surface and into the paste. As a result, a higher percentage of efficiency in the colour change of the MB solution is reached, Figure 6b.

It is possible to observe a combination of three different effects difficult to distinguish: (i) adsorption (affinity of MB molecules to be attracted to the porosity of the material mainly driven by electrostatic forces); (ii) photodegradation (degradation of MB molecules as a result of photoactivation of semiconductor nanoparticles); (iii) photolysis (effect of the lamp when it is on). Since AAM shows high basicity for a pH > 13, it is suggested that present a negative charge on the surface [25]. Therefore, the mechanism of photocatalytic reaction begins with the adsorption of MB molecules on the surface of the pastes due to presence of OH- groups, then, once the AAM are under UV illumination and the nanoparticles absorbing a photon on the surface an electron-hole pair is formed reacting with H_2_O molecules to form radicals OH· which oxidized to MB molecules. In the case of the metakaolin-based pastes, those with nanospheres present a better behaviour. These results are consistent with previous research where ZnO nanoparticles in geopolymer material have been used to performed photocatalytic test using MB [18] and are better than the results shown in [26] using ZnO nanoparticles and white cement, even when the concentration and irradiation conditions are different. The pastes with nanorods showed similar behaviour with and without functionalization, but the efficiency was poor concerning pastes with nanospheres.

In the case of slag-based pastes, Figure 7, it is possible to observe a good interaction when the lamp is on. This could be related by the fact that MB molecules present a good interaction with the nanostructure of C-A-S-H surface [27], where cation exchange of Ca^+^ for cationic monomers and dimer species onto anionic sites is done

No significant change in residual concentration of MB solution between samples with and without nanoparticles at the end of the test is observed.

Little difference in photodegradation between the nanoparticles with CMC and without is observed in Figure 6 and Figure 7. To identify the influence of the three different effects clearly, each one of them is present for the metakaolin-based pastes, Table 1, and metallurgical-based pastes, Table 2. The adsorption values are the average of numerous tests done in completely dark conditions. The percentage of photodegradation was determined subtracting to the total degradation under UV irradiation the value of adsorption average and the photolysis percentage. The effect of the CMC functionalization is more evident in the SL pastes than MK pastes. The reason is again the geopolymerization kinetics. Meanwhile, in MK pastes the geopolymerization process is very fast due to thermal treatment in which is cured, this provokes that the nanoparticles spend less time in basic conditions; in SL pastes, the curing conditions were at room temperature, and therefore, the nanoparticles spend more time exposed to highly basic conditions during geopolymerization. Previous research [28] had reported that zinc oxide is rapidly hydrolysed in the water at room temperature and at high pH (≈12–13.5) species as $\mathrm{Zn}{\left(\mathrm{OH}\right)}_{3}^{-}$ and $\mathrm{Zn}{\left(\mathrm{OH}\right)}_{4}^{2-}$ are dominant. Hence, the CMC functionalization avoids the nanoparticles’ breakdown, and therefore, better photocatalytic performance is observed when the nanospheres and nanorods were functionalized, see Table 2. A comparative of the values in the photodegrading percentage between nanospheres or nanorods functionalized and without CMC functionalization, better performance is observed in those functionalized with CMC, where the photodegradation percentage of the functionalized nanostructures, spheres and rods, is double for the SL-based pastes.

It is worth notice that in dark conditions, it is possible to observe some fluctuations of the absorbance. This behaviour could be related to the fact that the MB molecules absorbed on the surface of the AAM are desorbed due to a phenomenon of saturation of free sites and electrostatic repulsion effect of some cations such Ca^+^ and Na^+^ from the surface of the AAM. On the other hand, hydroxyl groups present on the surface of the geopolymer may attract and hold cationic organic species [29] as MB that generates the fluctuation observed.

According to these results, the efficiency of the pastes with nanoparticles functionalized (major to minor) follows the next order: mkSf > slSf > slRf > mkRf and with nanoparticles without functionalization the ranking is: mkS > slS > slR > mkR.

The best performance of the pastes is with the addition of functionalized nanospheres. This is mainly because the nanospheres have a higher surface area than nanorods, with are bigger nanoparticles. An increase in the surface area leads to an increase in the active sites on the nanoparticles surface and the numbers of reactive oxygen species such as radical hydroxyl and superoxide created on the surface of the catalyser are higher [30]. On the other hand, the pastes with the addition of nanorods show no significant effect. The results are ascribed to the own poor performance of large nanostructures resulting in low photodegradation activity. Finally, it could relate the photocatalytic performance of the SL-based pastes without nanoparticles to the content of TiO_2_ and Fe_2_O_3_ as semiconductors that could be activated under UV irradiation [18].

### 2.3. Adsorption and Photocatalytic Kinetics

The formation of the reaction products is a useful way to define the efficiency of new photocatalytic materials′ performance. In that way, the interaction of adsorption and photodegradation between the solid-liquid systems could be described mainly by two kinetic models, the pseudo-first-order model [31], Equation (3) pseudo-second-order model [6], and Equation (4).
(3)$$ln\frac{\left(q_{e}-\;q_{t}\right)}{q_{e}}=\;-k_{1}t$$
where q_e_ (mg/g) and q_t_ (mg/g) are the sorption capacities of MB dye at equilibrium and at a time t, respectively, and k_1_ (min^−1^) is the first-order rate constant. By plotting ln(C_0_/C_t_) versus t, k_1_ and q_e_ can be acquired from the slop and intercept, respectively.
(4)$$\frac{t}{q_{t}}=\;\frac{1}{k_{2}q_{e}^{2}}+\;\frac{1}{q_{e}}t$$
where k_2_ (g/(mg.min)) is the second-order rate constant. By plotting t/q_t_ versus t, the second-order rate constant (k_2_) and the equilibrium capacity (q_e_) are obtained from intercept and slope, respectively.

In the pseudo-first-model kinetic, it is assumed that the composition of AAM sample remains constant during all the tests while MB concentration decreases over time. In this model, the driving force to carry out the kinetics of adsorption processes is the difference between the concentration of the solute adsorbed at equilibrium, and the solute adsorbed concentration at a given time. The interactions are only physical; therefore, the MB molecules are attracted to the solid by electrostatic forces. In the pseudo-second-order model is assumed that one of the three reagents (MB-AAM-ZnO nanostructure) has a concentration smallest that the other two, in this case, the MB concentration is very low compared with the others. Moreover, the interactions are by chemisorption, and the sorption capacity is proportional to the number of active sites occupied on the sorbent [32].

Table 3 shows the parameters of both kinetic models used in the regressive analysis. It may be seen that the correlation coefficients (R^2^) for the linear regressive indicate a better fit for the pseudo-second-order rate equations for all samples. If it is compared to the samples’ values under UV irradiation between those with CMC modification and without it, higher values are observed in those with CMC modification. These results show a better chemical bond between absorbent and absorbate in a monolayer on the surface of the AAM, and this fact strengthens the idea that the superficial modification of the ZnO with CMC helps to protect the nanoparticles in basic environments.

Table 4 shows that the SL pastes’ experimental data fit the pseudo-second-order kinetics model in all the samples, except when it added the nanospheres without functionalization. Under UV light and in dark conditions, the parameters for these samples show similar results for both kinetic models, which indicated a competition between physical adsorption, usually by diffusion and chemisorption in the most external layers of the AAM.

The kinetic rate values increased, and the adsorption capacities are lower for the dark conditions concerning UV illumination conditions. This explains the similar behaviour for adsorption in dark and UV irradiation conditions. When the adsorption rate is higher, the adsorption capacity is small, which provokes a rapid saturation of active sites. On the other hand, when the adsorption rate is slow, the adsorption capacity is higher, which generates a better distribution of the MB molecules forming monolayers on available active sites.

Experimental data fit pseudo-second-order kinetic models for the metakaolin-based pastes and slag-based pastes, under UV irradiation and dark conditions, are shown in Figure 8 and Figure 9, respectively. It is observed from Table 4 and Table 5 that adsorption rate is promoted by UV irradiation, similar results had been reported previously for fly ash-based pastes [6].

## 3. Materials and Methods

### 3.1. Materials

This research used two different precursors: Metallurgical slag (SL) from ArcelorMittal steelmaking plant located in Lázaro Cárdenas, Michoacán State, México (before its use, the slag was ground for 24 h by high energy milling up to get particle size < 150 μm); and a Kaolin from Tisayuca, Hidalgo, México. The chemical composition (oxide weight percentage) of both materials was determined by X-ray fluorescence, Table 5. By DSC analysis, at 548 °C with a heating time of 120 min., a complete detected dehydroxylation of the starting kaolin and formation of metakaolinite occurred. To ensure full dehydroxylation the kaolin thermal treatment was carried out at 700 °C for two hours, until metakaolin (MK) was obtained as a precursor. The chemical activator used, sodium hydroxide (NaOH, ≥98%), was provided by Jalmek.

### 3.2. Preparation of the Alkali-Activated Materials

AAM samples were prepared by mechanically mixing stoichiometric amounts of the two-aluminosilicate precursors: Metakaolin (MK) -matrix prepared using metakaolin and sodium hydroxide solution, 8 M; metallurgical slag, (SL), matrix designed using metallurgical slag with sodium hydroxide solution, 4M. To obtain a good workability, the liquid/solid ratio (in weight) was established 0.8 for the MK matrix and 0.6 for the SL matrix. Then, mechanical mixed paste was vibrated during 120 s of, producing a homogenous slurry poured into cylindrical acrylic moulds of 10 mm × 5 mm, diameter and thickness, respectively.

The slurry was cured in a laboratory oven at 65 °C for 20 h for the MK matrix, while the SL matrix was cured at room temperature (21 °C) in a curing chamber with a relative humidity above 90% [33,34]. All the experiments were tested at the age of 28 days, and several measurements, an average value of three samples, were recorded to determine the values of photocatalytic and adsorption behaviour.

### 3.3. Addition of ZnO Nanostructures

Two different morphologies of ZnO nanoparticles were employed, nanospheres (S) with an average size of 38 nm and nanorods (R) with an average size of 190 nm. The ZnO nanoparticles were prepared using a polyol method [13] as follows: A certain amount of acetate dehydrate was dissolved in 10 mL of pure ethylene glycol by reflux three-neck flask at room temperature under vigorous stirring (400 rpm) and heated up to 160 °C. Deionized water injected dropwise, keeping a fixed hydration ratio, H, of 8, defined as H = H_2_O/Zn. Once reached, 160 °C, the mixture was continuously stirred for another five h. The ZnO obtained was dried in an oven at 80 °C. Nanoparticles added directly to the paste without functionalization (W) and with functionalization (F) -mixed with 2% of Sodium Carboxyl Methyl Cellulose (CMC) to avoid the nanoparticles’ breakdown. This procedure was made considering the basic pH of the paste to avoid the breakdown of the nanoparticle. Finally, the NPs were mixed, with the cement paste, in a proportion of 10% of weight (90AAM-10ZnO NPs) and poured into the moulds. The ZnO content was selected according to mechanical strength analysis performed, results not shown in this work, where, at lower content, the compressive strength is poor and an increase in the content did not show higher compressive strength. Similar results in the ZnO content have been reported previously [18], and this behaviour is related to the condensation process during the geopolymerization process at ZnO content lower than 10%.

### 3.4. Photodegradation Studies of Methylene Blue

The dye under investigation, Methylene Blue (MB), with a labeled purity of more than 90%, was obtained from Sigma–Aldrich (St. Louis, MO, USA) and used without further purification. Deionized water was used to make the dye solutions of the calibration curve and work concentration.

A stock solution prepared by weighing 0.01628 g of MB diluted in 500 mL of deionized water. Then from the stock solution, 8.36 mL were used to prepare 100 mL of a methylene blue solution at a concentration of 8.5 × 10^−6^ M and then poured into a 200 mL glass beaker (Pyrex) a circular AAM sample. The distance between the liquid surface and the lamp was 15 cm. Figure 10 shows the experimental setup.

The adsorption and photocatalytic degradation of the MB was made in the dark and under ultraviolet (UV) illumination, respectively. This method was chosen in the present work to demonstrate the effectiveness of the composites’ photodegradation but is not better than the others are. However, it is suitable for aluminosilicates materials at high pH because its colour is stable under these conditions. Since that AAM are efficient absorbers of organic dyes because their surface hydroxyl groups can attract and hold cationic organic species [25]. Hence, in the present work, it is possible to evaluate the photodegradation performance of the composites attributed to the combined mechanisms (adsorption and photodegradation under UV illumination).

The photodegradation measurements were carried out by mixing a cylindrical sample, 0.5 mg of the composites with 100 mL of the aqueous MB solution with magnetic stirring. For each measurement, a 4 mL aliquot of the MB solution was taken, and the volume centrifuged off. The dye concentration in the solution was determined by measuring the maximum absorbance at 664 nm in the UV–vis spectrum using an Agilent 8453 spectrophotometer (Santa Clara, CA, USA) operated in single beam mode at room temperature in the wavelength range of 200–1100 nm.

First, to determine the adsorption equilibrium time, the experiment was conducted in dark conditions until the MB concentration in the solution decayed to an equilibrium value. The equilibrium reached during the first 30 min. Then, the degradation was divided into two steps, adsorption in dark conditions and photodegradation under UV illumination. The sample was first held in the dark until the MB concentration in the solution decayed to an equilibrium value during the first 30 min, then the Philips Actinic BL 15 W/10 SLV fluorescent lamps emitting at 350 nm–400 nm (optimally at 360 nm), was switched on and the photodegradation reaction allowed to come to completion. All experiments were done in triplicate to ensure reproducibility of the methodology. The average values were taken and plotted into graphs.

The removal efficiency, η, of the dye was then calculated by Equation (5):(5)$$\eta \;\left(\%\right)=\;\frac{C_{0}-{\;C}_{t}}{C_{0}}\;\times \;100$$
where C_0_ and C_t_ are the concentration of the dye solution at the initial time and after time t of absorption.

### 3.5. Characterization Techniques

Mineralogical studies of the cementitious pastes were carried by X-ray diffraction (XRD) (Bruker model D8 Advance) with Ni-filtered Cu Kα radiation operating at 30 mA and 40 kV. Data recorded in the 5–60° 2θ range (step size 0.019732° and 0.5 s counting time for each step). AAM’s microstructural analysis was performed by “field emission scanning electron microscopy” (FE-SEM) (Hitachi model S-4800) using an accelerating voltage of 20 kV. Semi-quantitative analysis of different samples was obtained by energy-dispersive X-ray spectroscopy (EDS) (Bruker model QUANTAX Esprit 1.9) provided by a beryllium (Be) window. SEM observations were performed on the fracture of different samples. Samples were carbon-coated in a sputter (Balzers SCD 050) and characterized at 28 days.

## 4. Conclusions

New composites of inorganics matrix with photocatalytic ZnO nanoparticles (NPs) were obtained. According to the experimental results, the next conclusions can be drawn:The addition of nanometric size ZnO into the inorganic polymer matrices produces satisfactory photocatalytic results.Nanospheres show a better photocatalytic performance than nanorods, when both are mixed with the pastes. The photodegradation performance in SL-based pastes with a small percentage of nanoparticles functionalized with CMC showed that low kinetics of geopolymerization could help avoid the ZnO nanostructures’ breakdown.Functionalization of ZnO nanoparticles with CMC avoids its breakdown under highly basic pH conditions, increasing the photocatalytic performance.The MK matrix showed that the addition of ZnO nanospheres, functionalized and without functionalization, increased MB’s photodegradation more than without nanoparticles. The addition of nanorods showed a slight increase in the photocatalytic activity.In SL matrix, a little increase in the photodegradation of MB dye was observed when functionalized nanospheres were added. In the rest of the experiments, no significant results were obtained.The MB adsorption and photodegradation follow pseudo-second-order kinetics. In dark conditions, the adsorption effect was responsible for dye removal. Meanwhile, under UV irradiation, a combination of photodegradation and adsorption effects are responsible for the decrease in MB concentration.These new ZnO-based composites have an enormous potential for the removal of organic pollutants from wastewaters and important environmental applications.

## Figures and Tables

**Figure 1 molecules-25-05519-f001:**
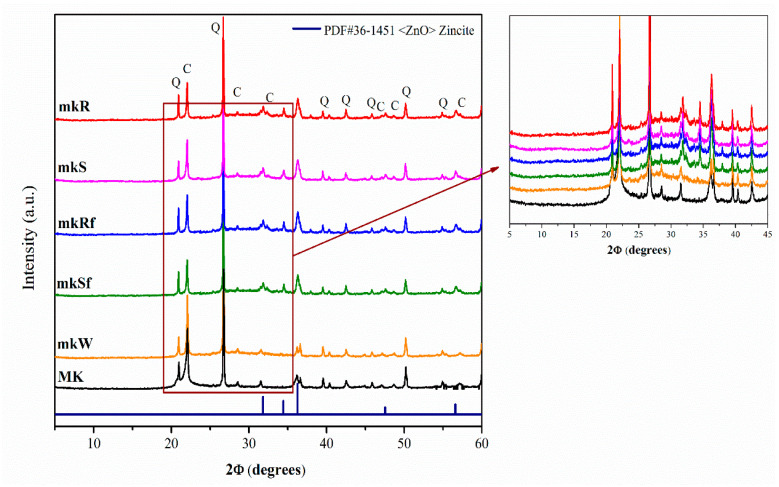
XRD diffractograms of metakaolin-based pastes with nanoparticles functionalized and without functionalization.

**Figure 2 molecules-25-05519-f002:**
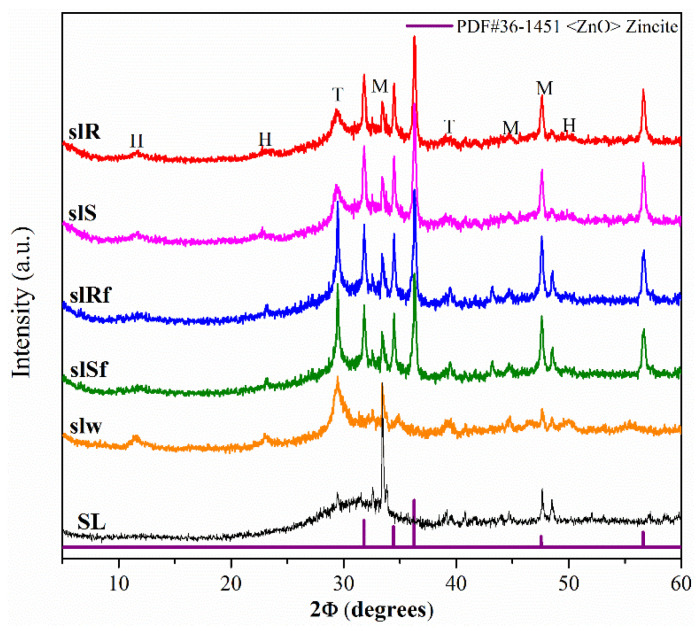
XRD diffractograms of metallurgical slag-based pastes with nanoparticles functionalized and without functionalization.

**Figure 3 molecules-25-05519-f003:**
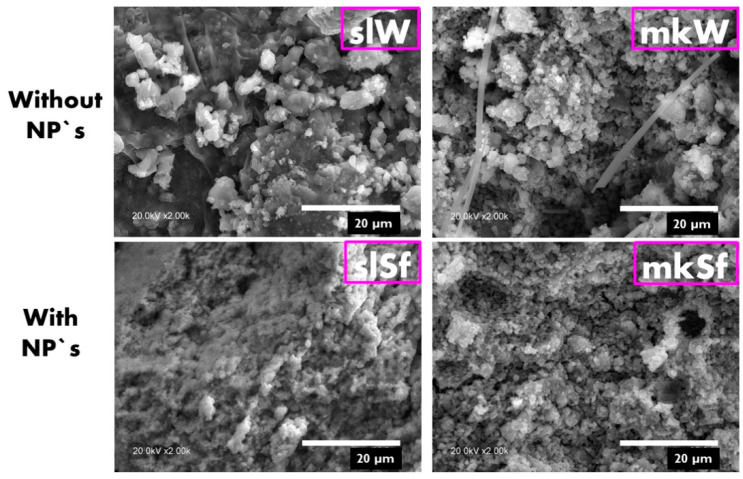
SEM images of the metallurgical slag (SL) and metakaolin (MK) pastes without and with nanoparticles.

**Figure 4 molecules-25-05519-f004:**
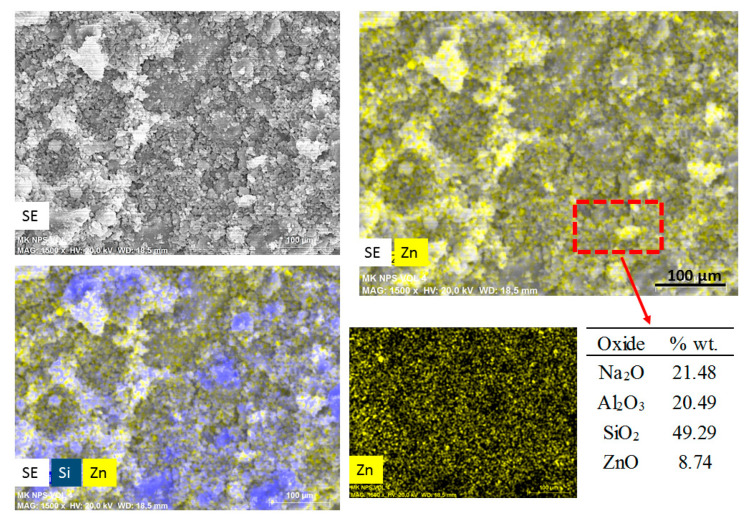
SEM images of secondary electros and EDS analysis for metakaolin-based paste with addition of nanospheres functionalized (mkSf).

**Figure 5 molecules-25-05519-f005:**
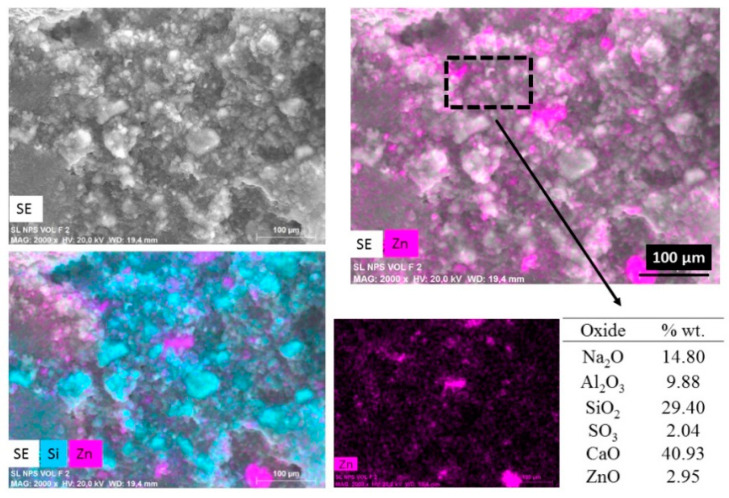
SEM images of secondary electros and EDS analysis for metallurgical slag-based paste with addition of nanospheres functionalized (slSf paste).

**Figure 6 molecules-25-05519-f006:**
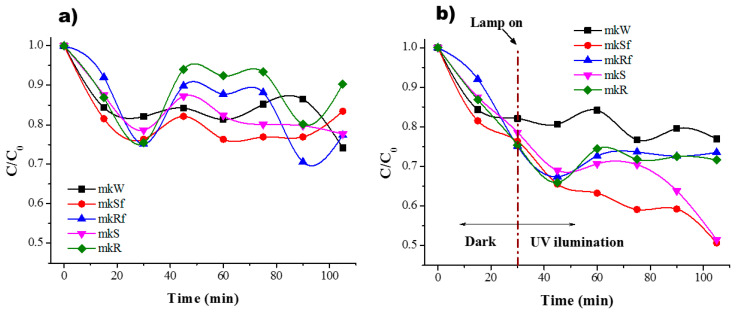
Residual concentration of MB (**a**) in dark conditions and (**b**) under UV irradiation for the MK paste and with ZnO nanostructures.

**Figure 7 molecules-25-05519-f007:**
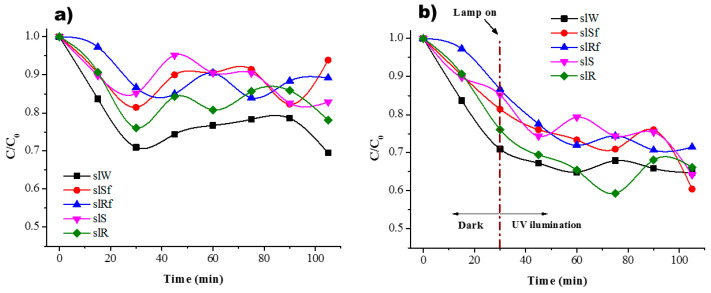
Residual concentration of MB (**a**) in dark conditions and (**b**) under UV irradiation for the SL paste and with ZnO nanostructures.

**Figure 8 molecules-25-05519-f008:**
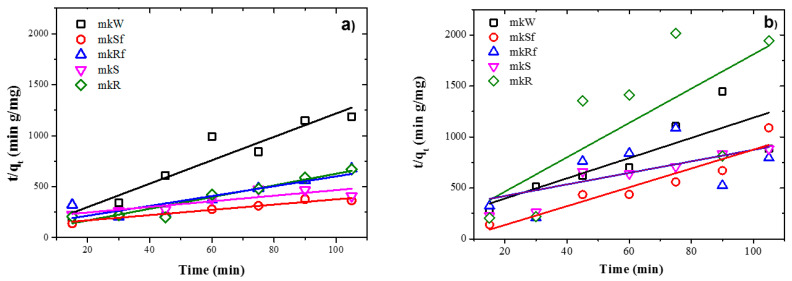
Plots of the pseudo-second order kinetics of MB adsorption on metakaolin-based alkaline cements with addition of ZnO nanoparticles in; (**a**) under UV irradiation and (**b**) dark conditions.

**Figure 9 molecules-25-05519-f009:**
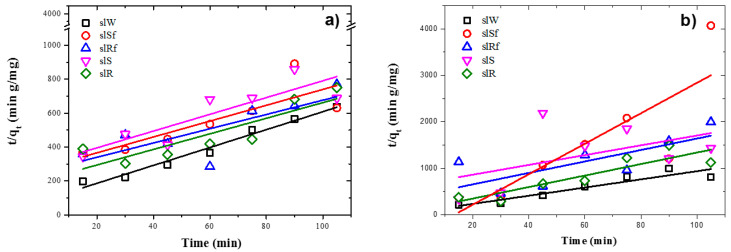
Plots of the pseudo-second order kinetics of MB adsorption on metallurgical slag-based alkaline cements with addition of ZnO nanoparticles in; (**a**) under UV irradiation and (**b**) dark conditions.

**Figure 10 molecules-25-05519-f010:**
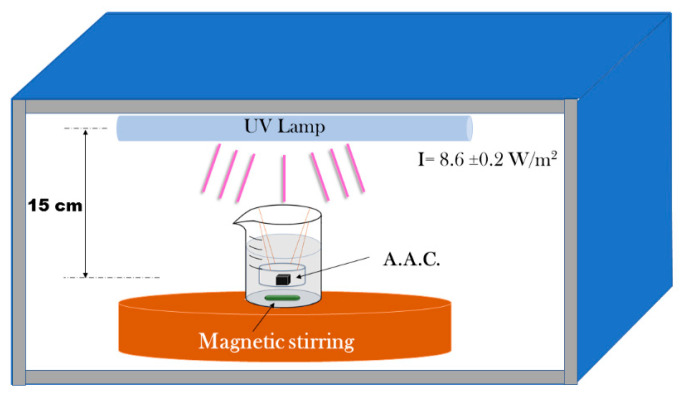
Experimental set up for the photocatalytic studies.

**Table 1 molecules-25-05519-t001:** Effects in the photodegradation process in metallurgical slag-based pastes.

Sample	Photolysis (%)	Adsorption (%)	Photodegradation (%)	Total Degradation (%)
Metakaolin				
mkW_Dark	0.00	25.85	0.00	25.85
mkW_UV	5.03	18.06	0.00	23.09
Nanospheres				
mkSf_Dark	0.00	16.59	0.00	16.59
mkSf_UV	5.03	16.59	27.71	49.33
mkS_Dark	0.00	22.35	0.00	22.35
mkS_UV	5.03	22.35	21.10	48.48
Nanorods				
mkRf_Dark	0.00	22.66	0.00	22.66
mkRf_UV	5.03	18.06	3.38	26.47
mkR_Dark	0.00	9.70	0.00	9.70
mkR_UV	5.03	18.06	5.24	28.33

**Table 2 molecules-25-05519-t002:** Effects in the photodegradation process in metallurgical slag-based pastes.

Sample	Photolysis (%)	Adsorption (%)	Photodegradation (%)	Total Degradation (%)
Metallurgical slag				
slW_Dark	0.00	30.22	0.00	30.22
slW_UV	5.34	30.22	0.00	35.56
Nanospheres				
slSf_Dark	0.00	11.42	0.00	6.14
slSf_UV	5.34	11.42	22.79	39.55
slS_Dark	0.00	17.14	0.00	17.14
slS_UV	5.34	17.14	13.33	35.81
Nanorods				
slRf_Dark	0.00	10.79	0.00	10.79
slRf_UV	5.34	10.79	12.39	28.52
slR_Dark	0.00	21.90	0.00	21.90
slR_UV	5.34	21.90	6.57	33.81

**Table 3 molecules-25-05519-t003:** Parameters of kinetics models for the adsorption of MB on metakaolin-based alkaline cement with and without ZnO nanoparticles.

Sample	Pseudo-First Order Parameter			Pseudo-Second Order Parameter		
	k_1_ (min^−1^)	q_e_ (mg/g)	R^2^	k_2_ [g/(mg·min)]	q_e_ (mg/g)	R^2^
Nanospheres						
mkW_UV	0.0003	723.000	0.0648	1.861	0.087	0.9126
mkW_Dark	0.0013	78.000	0.4709	0.495	0.101	0.6781
mkSf_UV	0.0049	30.592	0.9445	0.057	0.387	0.9485
mkSf_Dark	0.00004	5970.000	0.0013	1.791	0.108	0.8999
mkS_UV	0.0048	16.938	0.7584	0.039	0.364	0.8499
mkS_Dark	0.0010	14.180	0.4381	0.375	0.133	0.8801
Nanorods						
mkW_UV	0.0003	723.000	0.0648	1.861	0.087	0.9126
mkW_Dark	0.0013	78.000	0.4709	0.495	0.101	0.6781
mkRf_UV	0.0016	122.313	0.2702	0.1956	0.208	0.8105
mkRf_Dark	0.0016	60.250	0.2441	0.1034	0.177	0.3480
mkR_UV	0.0011	231.364	0.0964	0.6143	0.174	0.9175
mkR_Dark	0.0050	34.040	0.0438	2.2081	0.059	0.5298

**Table 4 molecules-25-05519-t004:** Parameters of kinetics models for the adsorption of MB on metallurgical slag-based alkaline cement with and without ZnO nanoparticles.

Sample	Pseudo-First Order Parameter			Pseudo-Second Order Parameter		
	k_1_ (min^−1^)	q_e_ (mg/g)	R^2^	k_2_ [g/(mg·min)]	q_e_ (mg/g)	R^2^
Nanospheres						
slW_UV	0.0022	109.545	0.6033	0.345	0.190	0.9782
slW_Dark	0.0007	333.571	0.1325	1.279	0.114	0.8770
slSf_UV	0.0034	25.147	0.6954	0.080	0.214	0.7755
slSf_Dark	0.0004	361.500	0.0448	2.462	0.031	0.6952
slS_UV	0.0030	27.000	0.7913	0.082	0.202	0.7505
slS_Dark	0.0008	97.500	0.2642	0.170	0.095	0.2538
Nanorods						
slW_UV	0.0022	109.545	0.6033	0.345	0.190	0.9782
slW_Dark	0.0007	333.571	0.1325	1.279	0.114	0.8770
slRf_UV	0.0033	26.485	0.2931	0.0032	0.664	0.5492
slRf_Dark	0.0006	146.333	0.1358	0.3698	0.081	0.5553
slR_UV	0.0032	51.7500	0.5695	0.1045	0.218	0.7664
slR_Dark	0.0005	320.800	0.0587	1.4725	0.081	0.7974

**Table 5 molecules-25-05519-t005:** XRF analysis (wt.%) of Tizayuca kaolin and metallurgical slag.

**Kaolin**	**SiO_2_**	**Al_2_O_3_**	**SO_3_**	**K_2_O**	**P_2_O_5_**	**TiO_2_**	**CaO**	**Fe_2_O_3_**	**ZnO**	**Others**	**LOI** **†**
	66.79	19.26	1.93	0.62	0.40	0.23	0.11	0.10	0.001	0.119	10.43
**Metallurgical Slag**	**CaO**	**SiO_2_**	**MgO**	**Al_2_O_3_**	**SO_3_**	**TiO_2_**	**Fe_2_O_3_**	**Na_2_O**	**K_2_O**	**Others**	
	44.76	30.52	10.20	9.59	2.51	0.74	0.61	0.38	0.35	0.35	

**†**. Loss of Ignition at 1000 °C.

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
