# Peer review of "ZnO Nanoparticles for Photocatalytic Application in Alkali-Activated Materials"

_molecules, 2020, doi:10.3390/molecules25235519_

Round 1

Reviewer 1 Report

The subject of manuscript is interesting. So the manuscript is suitable for publication. But there are some points which should be clarified before publication. I recommend minor revision of the manuscript:

  1. Although the information provided is very interesting, it is not clear why Methylene Blue was chosen for this study.

  1. Lot of work has already been done on ZnO nanoparticles for photocatalytic application. How the proposed method is better than the existing methods for this purpose ?

  1. There is no comparison of the results obtained by Authors with the results published previously.

  1. Line 63:MK… and Line 68:XRD…, abbreviations should be defined at first mention and used consistently thereafter.

  1. The Figure 3 of this manuscript should be discussed clearly.

  1. Line 253: The volume units 'l' or 'ml' should be replaced by 'L', 'mL'.

  1. Line 360: The author should check all the references according to the author guide.

Author Response

Title: ZnO nanoparticles for photocatalytic application in alkali-activated materials

We greatly appreciate Reviewers #1 and #2 for their valuable comments and suggestions. For the convenience of the Reviewers and Editor, we addressed these comments and suggestions point by point below, and the corresponding changes were highlighted in red in the revised manuscript.

Reviewer 1

1-Although the information provided is very interesting, it is not clear why Methylene Blue was chosen for this study.

Response: We thank the Reviewer for the important question.

The cationic dye, Methylene Blue, was chosen due to its reflected adsorptive capacity which is higher than other organic dyes [23]. It is worth mentioning the relationship between the adsorption and the photocatalytic activity since the organic dyes' photodegradation reaction occurs after adsorbed on the surface [24]. As a result, adsorption contributes simultaneously to accelerate photocatalytic action and participate in removing dyes. The MB colour is also stable at the high pH of the Alkali Activated Materials (AAM), [17].  Hence, in the present work, MB is considered suitable to evaluate the composites' photodegradation performance.  

2-Lot of work has already been done on ZnO nanoparticles for photocatalytic application. How the proposed method is better than the existing methods for this purpose?

Response:  We thank the Reviewer for the positive comment and its question:

The method is chosen in the present work to demonstrate the effectiveness of the composites’ photodegradation but, is not better than the others. However, it is suitable for aluminosilicates materials at high pH. Since these are efficient absorbers of organic dyes because their surface hydroxyl groups can attract and hold cationic organic species [25]. Hence, in the present work, it is possible to evaluate the photodegradation performance of the composites attributed to the combined mechanisms (adsorption and photodegradation under UV illumination). 

3-There is no comparison of the results obtained by Authors with the results published previously.

Response:  We thank the Reviewer for the positive comment and its question:

Due to the few research of photocatalytic applications of ZnO nanoparticles into alkaline activated materials, it is hard compared our result with literature. Nevertheless, our results are consistent with previous and recent research where ZnO nanoparticles in geopolymer material have been used to performed photocatalytic test using MB [18] and are better than the results showed in [26] using ZnO nanoparticles and white cement, even when the concentration and irradiation conditions are different, which makes a complicated comparison.

4-Line 63:MK… and Line 68:XRD…, abbreviations should be defined at first mention and used consistently thereafter.

 Response: We thank the Reviewer for the comment and the abbreviations have been properly defined.

5-The Figure 3 of this manuscript should be discussed clearly.

 Response: We thank the Reviewer for the positive comments and the Figure 3 has been discussed as follows:

SL-bases paste doped with functionalized ZnO NPs lead some changes in the microstructure, showing an increase in the compactness, as shown in Figure 3. On the contrary, in the MK-based pastes, no significant differences in the microstructures are observed.

6- Line 253: The volume units 'l' or 'ml' should be replaced by 'L', 'mL'.

Response:  We thank the Reviewer for the positive comments. The volume units 'l' or 'ml' have been changed to ‘L', 'mL' according to the Reviewer`s comment. 

7-Line 360: The author should check all the references according to the author guide.

Response: We thank the Reviewer for the positive comments and the references have been checked and modified according to the author guide.

Reviewer 2 Report

The authors report on the incorporation of ZnO nanoparticles in alkali-activated materials (AAM) for the development of efficient photocatalytic composites. The AAM samples were prepared by using two different aluminosilicate precursors, metakaolin and a metallurgical slag, and were mixed with ZnO nanospheres and nanorods. The composites and their constituents were characterized by XRD, SEM-EDX analysis and their photocatalytic activity was evaluated on methylene blue degradation under UVA illumination.  Overall, the work is well-aimed and systematic presenting interesting results on photocatalytic AAM composites from industrial wastes. However, before publication the following points should be considered

1) How was the ZnO content selected? Was there any optimization performed? Does the ZnO nanoparticles size or crystal quality play any role in the different photocatalytic performance.

2) The dark MB adsorption curves in Figures 6a and 7a show some fluctuations with time that do not always show a tendency to stabilize and thus reach equilibrium. This behavior should be discussed.

3) Are there any data on the materials surface area? Is the AAMs surface area expected to contribute in the different adsorptive/photocatalytic efficiency?

Some minor points:

Figure 3 should be cited and discussed in the manuscript.

line 53: “Superficial” may be better replaced by surface and not be used two times.

Author Response

Title: ZnO nanoparticles for photocatalytic application in alkali-activated materials

We greatly appreciate Reviewers #1 and #2 for their valuable comments and suggestions. For the convenience of the Reviewers and Editor, we addressed these comments and suggestions point by point below, and the corresponding changes were highlighted in red in the revised manuscript.

Reviewer 2

1) How was the ZnO content selected? Was there any optimization performed? Does the ZnO nanoparticles size or crystal quality play any role in the different photocatalytic performance.

Response: We thank the Reviewer for the positive comments.

The ZnO content was selected according with mechanical strength analysis performed, results not shown in this work, where at lower content the compressive strength is poor and an increase in the content did not show higher compressive strength. Similar results in the ZnO content are reported previously [18] and this behaviour is related to the condensation process during geopolymerization process at ZnO content lower of 10 %. On the other hand, a better photocatalytic performance of the AAM with nanospheres is related mainly to higher surface area than the nanorods which are bigger nanoparticles with respect to nanospheres. In addition, an increase in the surface area leads to an increase in the active sites on the nanoparticles` surface and the number of reactive oxygen species such as radical hydroxyl and superoxide created on the surface of the catalyser are higher [30]. The pastes with the addition of nanorods show no significant effect. The results are ascribed to the own poor performance of large nanostructures resulting in low photodegradation activity.

2) The dark MB adsorption curves in Figures 6a and 7a show some fluctuations with time that do not always show a tendency to stabilize and thus reach equilibrium. This behaviour should be discussed.

Response: We thank the Reviewer for the positive comments

In dark conditions it is possible to observe some fluctuations of the absorbance, this behaviour could be related to the fact that the MB molecules absorbed on the surface of the AAM are desorbed due to a phenomenon of saturation of free sites and electrostatic repulsion effect of some cations such Ca+ and Na+ from the surface of the AAM. On the other hand, hydroxyl groups present on the surface of the geopolymer may attract and hold cationic organic species [29] as MB which generates the fluctuation observed.

3) Are there any data on the materials surface area? Is the AAMs surface area expected to contribute in the different adsorptive/photocatalytic efficiency?

Response: We thank the Reviewer for the positive comments

Not surface area analysis was performed for the two different AAM materials. This work is part of a more extensive research in order to a better understanding of the mechanisms involve in the photocatalytic and self-cleaning behaviour caused by the addition of ZnO nanoparticles in alkaline activated materials. In that way, we are performed a more exhaustive characterization including textural properties of the composites and identify the effect of the surface area in the adsorptive/photocatalytic performance in this kind of materials.

 Some minor points:

Figure 3 should be cited and discussed in the manuscript.

Response: We thank the Reviewer for the positive comments and the Figure 3 has been discussed as follows:

SL-bases paste doped with functionalized ZnO NPs lead some changes in the microstructure, showing an increase in the compactness, as shown in Figure 3. On the contrary, in the MK-based pastes, no significant differences in the microstructures are observed.

line 53: “Superficial” may be better replaced by surface and not be used two times.

Response:  We thank the Reviewer for the positive comments and the “superficial” word have been delated and the paragraph was modified without change the idea.